# Politics in Search of Evidence—The Role of Public Health in the COVID Pandemic in Germany: Protocol for a Situational Analysis

**DOI:** 10.3390/ijerph192416486

**Published:** 2022-12-08

**Authors:** Julia Piel, Marilena von Köppen, Christian Apfelbacher

**Affiliations:** Institute for Social Medicine and Health Systems Research, Medical Faculty, Otto von Guericke University, 39120 Magdeburg, Germany

**Keywords:** public health, politics, COVID-19, evidence-based knowledge

## Abstract

The protocol presents a research project that explores the relationship between science and politics in the COVID-19 pandemic from the perspective of public health (PH) scientists in Germany with situational analysis (SA). In context of this global crisis, we ask how PH scientists negotiate their roles as scientists and political citizens; how PH scientists perceive the relationship between their own and other scientific disciplines; and which normative assumptions PH scientists make in the production and dissemination of research findings. To conduct SA, we combine qualitative interviews with PH experts and published documents from scientific societies in PH and related disciplines (e.g., position and opinion papers) to analyze the complexity of integrating evidence-based knowledge into politics. Data are analyzed using different SA mapping tools, focusing on social worlds/arena maps and positional maps. The approach will reveal both explicit positions in the PH community and implicit or hidden voices and will reflect normative assumptions as well as internal structures of PH in Germany. The findings will be discussed with the philosophy sub-project and in a stakeholder workshop with politicians and the public. Further insights will be gained for politics and PH responses to future global crises.

## 1. Introduction

In view of the variety of global problems that are characterized by a high degree of complexity, an intensive discourse has emerged in recent decades on what contribution sciences can make in the political response to these problems. This applies equally to financial, climate and health politics. It is controversial how politics should adequately process scientific findings and what political consequences result from this, for example, for the understanding of democracy [1,2]. The COVID-19 pandemic has once again highlighted the importance of this question. 

Habermas distinguishes three models of the relationship between politics and science, namely a technocratic model, a decisionist model and a pragmatic model [3,4] (pp. 120–145). While the technocratic model pleads for a technical-bureaucratic implementation of evidence-based knowledge and the decisionist model postulates a priority of political decision-making over scientific knowledge, the pragmatist model argues for an integrative process of science and politics and a mutual respectively experimental learning process [5,6,7,8]. The following questions have been subject to further elaboration: Under which conditions can these learning processes thrive, and how can policy advice be shaped in the face of deeply contested issues? Further elaboration has led to a plea for a “well-ordered science” [9], a “pragmatic-enlightened” model of assessment making [3] or the idea of scientists as “honest brokers” [10]. Engaging in such learning processes is especially important because of the high complexity and dynamics of the pandemic situation, which affects not only the health of the people but almost all social, economic or cultural dimensions of individual and social life [11,12]. 

Regarding the different roles of politics and science, politics faced the need to adopt invasive policies that are highly normatively charged (see, for example, the discourses around lockdowns or vaccinations framed as the antithesis of personal freedom vs. health protection) [13,14]. Since the beginning, policymakers have therefore asked for evidence-based knowledge to guide political decision-making [15,16,17]. Public health—as an umbrella discipline combining knowledge from epidemiology to economics, sociology, psychology, cultural sciences and ethics—is especially apt to provide the necessary insights [18].

In Germany, however, policymakers preferred the knowledge of virology, infectology and critical care medicine on their scientific advisory boards [19,20]. These expertises sought to provide scientific legitimacy for the political implementation of population-wide interventions, such as restrictive social distancing measures, even if evidence was not (yet) available [21,22]. The rationale for emphasizing biomedical scientific perspectives was the underlying normative intent of preventing a collapse of the health system leading to the tragic death of many infected people—a scenario which happened in some of the worst affected countries [23,24]. While scientists acknowledged this goal, they also voiced the concern to consider social consequences of interventions, e.g., for certain social groups or people in certain locations, such as nursing homes or reception centers and asylum camps [25,26]. However, the focus on biomedical perspectives in policy advice was maintained.

At the same time, public health scientists became increasingly organized in scientific networks (e.g., scientific societies) to strengthen the voice of their disciplines in public debates and to assume their role of providing policymakers with the necessary scientific expertise in response to the COVID-19 pandemic. They released several position papers and policy briefs offering their perspectives on the pandemic and their recommendations for policy action [27]. Some of these papers are explanatory in nature (e.g., explaining findings from modeling studies); others generated evidence during and after previous SARS and MERS outbreaks. In a number of the papers, (strong) recommendations were given. The guiding normative assumptions, however, were rarely analyzed and critically reflected in everyday scientific practice. 

This protocol presents a research project that aims to reflect normative assumptions in science-policy integration on the COVID-19 pandemic in Germany and to clarify the position of public health (PH) in this situation. “Politics in Search of Evidence“ (PoSEvi) is a research alliance between PH and practical philosophy. The collaboration consists of two subprojects. The study protocol focuses on the PH subproject located at the Institute of Social Medicine and Health Systems Research at the Otto von Guericke University Magdeburg. The team of the philosophy subproject affiliates with the Munich School of Philosophy. 

In our research, we focus on perspectives of PH scientists and examine how they experience the relationship between evidence-based knowledge and political decision-making in the context of the COVID-19 pandemic. 

The objectives of the project are to explore and understand: How PH scientists perceive the relationship between their discipline and political decision-making;How PH scientists negotiate their own role as scientists and political citizens;How PH scientists perceive the relationship between their scientific discipline and others;The normative assumptions PH scientists make in the production and dissemination of research findings;How political and cultural contexts influence the self-image of PH scientists and their relationship to politics.

## 2. Materials and Methods

### 2.1. Study Design

The study employs a multi-method approach (see Figure 1): semi-structured interviews with experts, document analysis and stakeholder dialogue. We will conduct reflexive interviews with representatives from the disciplines of PH sciences in Germany to explore and understand their subjective perspectives. The interview data will be complemented with documents published by most relevant German scientific societies during the pandemic. Both types of data will be combined in a situational analysis [28].

### 2.2. Situational Analysis

To analyze the complex embeddedness of normative positions in science and politics on the pandemic, we draw on the theoretical approach of social world/arena, as developed in the context of Adele Clarke’s situational analysis (SA) [29].

The SA builds on the tradition of the Grounded Theory (GT) and broadens the analysis perspective by focusing on the situatedness of the phenomena under study [29]. “SA first and foremost seeks to understand the dense complexities of a particular situation broadly conceived” [29] (p. xxiv). In this context, the situation is not synonymous with the concept of context: “The word context clearly denotes that which surrounds something, but assuredly is not part of it”, but in SA it is assumed that “entities in relation to each other are constitutive of each other” (p. 17). This “co-constitutiveness” and thus the question “what co-constitutes what else in a given situation” is the core concern of SA [29] (p. 18). SA relies on the social world/arena framework referred to Strauss [30] but is further developed: Social worlds are multiple collective actors sharing common perspectives “that form the foundation for both individual and collective identities and for commitment to collective action” [29] (p. 148). 

In the situation of the COVID-19 pandemic, we encounter many different social worlds. Some of them existed already before (e.g., as the various scientific societies); others have only emerged during the pandemic (e.g., the German Competence Network Public Health for COVID-19). Moreover, public health scientists do not only belong to professional social worlds but are also part of worlds that are more private in nature (family, neighborhoods, school, clubs). With their respective logics, norms and orientations, these different social worlds engage in discourses in social arenas, where “various issues are debated, negotiated, fought out, forced and manipulated by representatives” of the participating worlds and subworlds [31] (p. 124). Social arenas, therefore, usually consist of several social worlds actively pursuing their agendas, e.g., by normative positioning. The question of what exactly is being argued about in a specific social arena and how these are demarcated from one another can ultimately only be determined by the analysis and research focus. For our study, social arenas can be defined as the negotiation of the relation between scientific knowledge from different disciplines and politics. Depending on the aspect of interest, more abstract (e.g., an arena of policy deliberation) or more concrete tailoring (e.g., an arena in which the issue of school closures is negotiated) can be chosen for this purpose.

### 2.3. Interviews with Experts

Scientists with at least master-level qualification will be approached through different channels (e-mail, telephone, Twitter), drawing on the established scientific networks of the researchers. Participants will also be recruited via the snowball principle [32]. Although PH scientists are concerned with the study of the distribution and determinants of health-related states and events in specified populations, as well as the application of this study to the control of health problems [33], there are fine differences between subdisciplines (e.g., focus on infectious or chronic disease, clinical epidemiology, considering social and environmental determinants), whether research is methodological (e.g., modelling) or applied (e.g., social medicine) and whether scientists work in a medical or social science environment. Sampling will therefore be purposive to ensure the greatest possible heterogeneity regarding these differences and also regarding individual characteristics (e.g., age, gender, degree of seniority). Sampling will continue until theoretical saturation will have been reached, which is expected to happen after about 20–25 interviews.

### 2.4. Participants’ Recruitment and Informed Consent

We will contact potential respondents by email to participate in a semi-structured interview on the relationship between science and policy during the COVID-19 pandemic. At the same time, participants will receive a letter with information about the aims and research objectives, implementation and processing of personal data in the project. Any questions will be clarified afterwards. Respondents will be required to sign an informed consent to participate in the study. A trusted third party at the Medical Faculty of Otto von Guericke University Magdeburg will collect the informed consent documents. The interview participants will be informed about the protective measures taken with regard to their data both during transcription and analysis. In particular, the material will be anonymized and text passages will be adapted to ensure that no identification of the participants will be possible. After the interviews, short summaries of themes derived from the interviews will be sent to the participants with a request for correction or completion (respondent validation) [34].

### 2.5. Document Analysis

The interview study is accompanied and deepened by a document analysis [35,36] of position and opinion papers, fact sheets and policy briefs published by relevant German PH societies, in which PH scientists are organized. We limit the document analysis to the scientific societies that existed prior to the pandemic because we consider the debates that take place there as the materialized discourse of the community. The different societies also represent the different normative assumptions of the PH subdisciplines. Thus, an analysis of the subworlds in the social arena becomes possible. The relevant professional societies are: German Society for Social Medicine and Prevention (DGSMP)URL: https://www.dgsmp.de/ (accessed on 1 November 2022);German Society for Medical Sociology (DGMS)URL: https://www.dgms.de/ (accessed on 1 November 2022);German Society for Public Health (DGPH)URL: https://www.dgph.info/ (accessed on 1 November 2022);German Society for Epidemiology (DGEpi)URL: https://www.dgepi.de/ (accessed on 1 November 2022);German Society for Medical Informatics, Biometry and Epidemiology (GMDS)URL: https://www.gmds.de/ (accessed on 1 November 2022).

We include all documents published by these societies as they relate to the COVID-19 crisis. So far, we identified 40 relevant documents. Papers published from political authorities or parties will not be considered for the purpose of this project.

### 2.6. Data Collection, Data Processing and Data Storage 

Interviews will be conducted using questions from a topic guide via a video conferencing system that complies with European data privacy regulations and is recommended by our institution. The recording will be done either by voice recorder or directly by the video conferencing system. Information will also be collected on age, years of professional activity, research or vocational focus, professional biography and membership in scientific societies. The interview questions refer to the personal perspective of the participants on the relationship between science and politics. We assume that focusing on the subjective viewpoint might be challenging for the participants, who may be more accustomed to being addressed as experts in the context of the pandemic. This can lead to irritations in the interview, which will be considered and managed by us. This is why we define the type of interview as “reflexive interview”, which enables a flexible use of the interview guide and the creation of a trustworthy interview atmosphere. Although the interviewer remains neutral throughout, questions that are philosophically grounded and immanent in the situation support cognitive reflection of the participants, similar to the think-aloud approach [37].

The interview guide is organized into three main topics. At the beginning, participants will be asked about their self-perception/role as representatives of their disciplines and about the responsibilities and tasks they carried out during the pandemic. The second thematic section includes questions about personal values and expectations in relation to political deliberation and decision-making processes during the pandemic. The aim is to obtain positions from role-specific standpoints, both as a scientist and as a private person. A third focus will be on their estimation of successful policy advice in times of crisis. We are interested in learning more about the importance they attach to evidence in advising policymakers through science. The transitions between the topics in the interview will be moderated appropriately for the context. Modifications of the interview guide, both during the research process and during the interview situation, are intended.

Interviews will be transcribed verbatim and pseudonymized. The original data (audio files) will initially be stored on a password-protected server of the Institute of Social Medicine and Health Systems Research. Only members of the PoSEvi-project will be explicitly authorized to access data of this study. After transcription, the audio files will be deleted immediately conforming to data protection guidelines. The transcribed data will also be stored separately from the written consent documents.

## 3. Data Analysis

Data will be analyzed using the software MAXQDA by combining GT-coding and SA-mapping to enable us to consider the intertwining discourses and actors in the arena of science-politics relationship during the complex and dynamic situation of the pandemic from the perspective of PH scientists [28,38]. The iterative process of coding, ordering and mapping the data will be performed by two researchers to ensure inter-subjectivity [35,39].

As analytical tools, SA provides four different mapping methods (1. situational map; 2. relational map; 3. social worlds/arenas map (Figure 2); 4. positional map (Figure 3)) [40]. While the first two methods are used to break down the data, social worlds/social arenas maps and positional maps allow the ecology of the situation to be represented. “Social Worlds/Arenas Maps lay out all of the collective actors and the arena(s) of commitment within which they are engaged.” [40] (p. 14). This allows us to capture and contextualize the different social, organizational and institutional dimensions of the situation. Using the positional maps, it is further possible to represent the full range of positions “taken and not taken in the discursive data found in the situation.” [40] (p. 15). For this purpose, the positions are plotted on key analytic axes [41]. Special attention is paid to positions that are missing in the data. 

Findings from the interviews with public health experts and the document analysis will be triangulated to generate a comprehensive theoretical understanding of normative positions among public health subdisciplines, other scientific disciplines and political decision-making in their cultural context.

## 4. Cooperation

Empirical findings from the SA conducted within the PH-project will be theoretically reflected and discussed with the philosophical project. Interdisciplinary dialogue will enhance the interpretation of the data on the complex relationship under research and eventually will enable a critical reflection of the concept of evidence-based policy. Finally, the reflections on scientific advice of politics from the perspective of PH scientists will be fed into a high-level stakeholder dialogue. To this dialogue forum specialists from politics, PH service, the medical sector and civil society will be invited to discuss the different scenarios of the relationship between sciences and politics in crisis situations. Referring to the pragmatic method of knowledge production, an iterative deliberation process will frame the dialogue forum. As a result, the forum will present various practical arguments concerning the (theoretical) scenarios of the science-politics collaboration and how different types of knowledge could be implemented in politics facing future pandemics and further global crises.

The PH subproject plans two publications on empirical findings and the methodological approach in scientific journals with PH reference. Further, we plan, together with the philosophical project, a joint popular science publication, which addresses a public readership.

## 5. Conclusions

The strengths of the study lie in its particular methodology. To the best of our knowledge, no study to date has examined the role of PH in the COVID-19 pandemic using SA. However, we believe an analysis, which considers the pandemic in its specific situatedness, to be very insightful. Precisely, because the various actors and elements are examined in all their complexity and interconnectedness rather than in isolation, the different relations between the various social worlds and sub-worlds in the social arena of pandemic governance can be uncovered. This applies especially when studying implicated actors, e.g., actors silenced or marginalized. The use of SA’s distinctive mapping analytics also helps to make visible the different normative assumptions and implicit settings in the field. Thanks to the intuitive experience of the different situational and positional maps, these insights can be fed back into the discourse with the philosophical subproject and in this way can strengthen the interdisciplinary collaboration [40].

It will be challenging to adapt our approach to meet the dynamics of the situation. The pandemic can be described in different phases, for example, the period before and after the availability of a vaccine. To examine how positions evolved over time and might evolve in the future, we follow the suggestion of Meszaros et al. [41] and will develop multiple maps on different events during the pandemic to capture the transformative nature of the situation.

## Figures and Tables

**Figure 1 ijerph-19-16486-f001:**
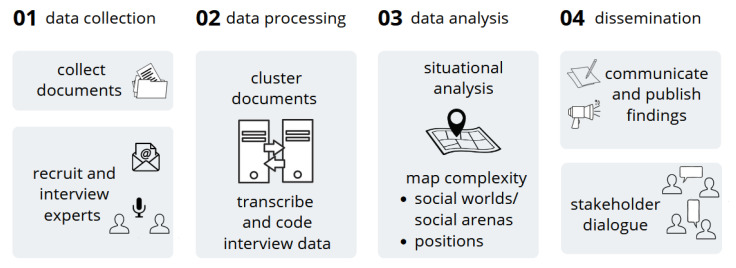
Research process in public health subproject.

**Figure 2 ijerph-19-16486-f002:**
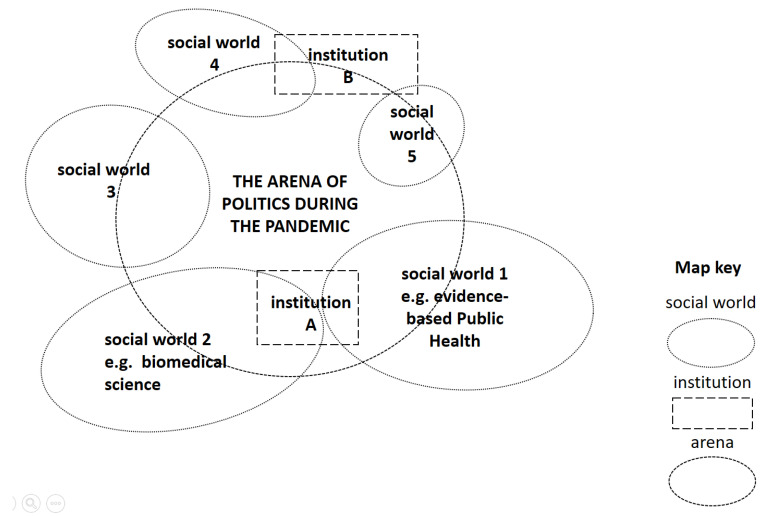
Exemplary social world/social arena map, own illustration referring to SA-template provided by SAGE Companion Website, Copyright © 2018 by SAGE Publications, Inc. Reprinted by permission of SAGE Publications, Inc. [42].

**Figure 3 ijerph-19-16486-f003:**
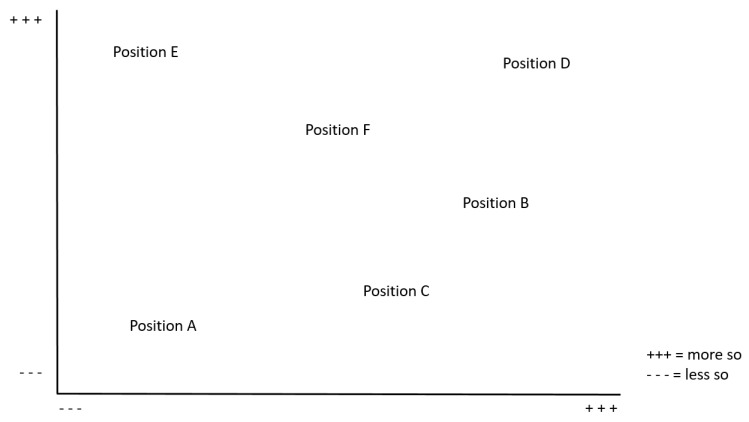
Abstract positional map, own illustration referring to SA-template provided by SAGE Companion Website, Copyright © 2018 by SAGE Publications, Inc. Reprinted by permission of SAGE Publications, Inc. [42].

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
