# Peer review of "Politics in Search of Evidence—The Role of Public Health in the COVID Pandemic in Germany: Protocol for a Situational Analysis"

_ijerph, 2022, doi:10.3390/ijerph192416486_

Round 1

Reviewer 1 Report

The authors present a very convincing and interesting research protocol on their ongoing research project on the role of public health experts between science and politics in the Corona Pandemic in Germany. I recommend publishing after minor revisions have been taken out:

Here are my recommnedations:

1. Introduction

I like the design of the project and its research topics, nevertheless I think the reader should get more information about why the questions 1-5 are important for scientific investigation. I think the authors should present briefly some information about current theoretical discussions about science-policy interactions and why it is important to distinguish between different roles from scientists and political actors. Here, literature on science-policy interfaces from political science and/or sociology could help.

2.4 document analysis

The authors should explain how these professional societies have been selected and especially why are not as well interdisciplinary societies or other kind of networks are selected? I think , e.g. LEOPOLDINA could be also important since its suggestions got some attention and here also public health experts were involved. All in all, the selection of these societies and non-selection of interdisciplinary networks need justifying.

Clear terminology

Please mind some terms not stemming from your discipline: politics/ policy. In my view sometimes it has to be re-checked if "policy" might be a better term than "politics". In political science "politics" mean the conflicts and debates in the political process, e.g. between different parties or interest groups. "Policy" means the contents: which problem is regulated, which instrument is used etc... Normally, science informs policy (evidence-based policy) but science-related solutions are controversially debated in politics. Health policies would be about the contents (e.g. mandatory vaccination , lockdowns etc.) whereas health politics would be about political controversies in the political process (e.g. mandatory masks or not (liberty vs. strong state intervention), different health policy concepts from CDU and SPD etc.)

I would say that in most cases "politics" is correct, but in Line237 it has to be "evidence-based policy" instead of "politics". (see as well the title from the  book from Cairney you cited).

minor issues:

L 49: it has to be "scientists" instead of "scientist"

L 53 please delete "and networks" in brackets or change the sentence, since "and networks" is written two times.

Author Response

Dear reviewer,

we gratefully appreciate your effort and attention to our manuscript. Your comments were very helpful in revising the text. We address each comment point-to-point in the following. The line numbers refer to the revised manuscript (see document manuscript_v.2_clean).

Best regards.

Point 1:

  1. Introduction

I like the design of the project and its research topics,  nevertheless I think the reader should get more information about why the questions 1-5 are important for scientific investigation. I think  the authors should present briefly some information about current  theoretical discussions about science-policy interactions and why it is  important to distinguish between different roles from scientists and  political actors. Here, literature on science-policy interfaces from  political science and/or sociology could help.

Response 1

Thank you for your constructive comment. We followed the advice to deepen the theoretical basis of the science-policy interaction by referring to the different models proposed by Habermas and subsequent developments by Kitscher and Pielke (see line 35-47). We also differentiated the roles of scientists and policy-makers (see line 48-51 and 67-70).

Point 2:

2.4 document analysis

The authors should explain how these professional societies have been  selected and especially why are not as well interdisciplinary societies  or other kind of networks are selected? I think , e.g. LEOPOLDINA could  be also important since its suggestions got some attention and here  also public health experts were involved. All in all, the selection of  these societies and non-selection of interdisciplinary networks need  justifying.

Response 2

The reviewer raises an important point here. We are  specifically interested in scientific societies and networks formed by public health scientists. Leopoldina did make important suggestions and produced valuable documents but there is only very little public health representation. We therefore limit the document analysis to the existing scientific societies because we consider the debates that take place there as the materialized discourse of the community. The different societies also represent the different normative assumptions of the public health sub-disciplines. Thus, an analysis of the sub-worlds in the social arena becomes possible (see line 173-191).

Point 3:

Clear terminology

Please mind some terms not stemming from your discipline: politics/  policy. In my view sometimes it has to be re-checked if "policy" might  be a better term than "politics". In political science "politics" mean  the conflicts and debates in the political process, e.g. between  different parties or interest groups. "Policy" means the contents: which  problem is regulated, which instrument is used etc... Normally, science  informs policy (evidence-based policy) but science-related solutions  are controversially debated in politics. Health policies would be about  the contents (e.g. mandatory vaccination , lockdowns etc.) whereas  health politics would be about political controversies in the political  process (e.g. mandatory masks or not (liberty vs. strong state  intervention), different health policy concepts from CDU and SPD etc.)

I would say that in most cases "politics" is correct, but in Line237  it has to be "evidence-based policy" instead of "politics". (see as well  the title from the  book from Cairney you cited).

Response 3

We have reviewed the use of the terms policy and politics and adjusted them accordingly (see line 264).

Point 4:

Minor issues:

L 49: it has to be "scientists" instead of "scientist"

L 53 please delete "and networks" in brackets or change the sentence, since "and networks" is written two times.

Response 4

Thank you very much, we addressed the minor issues.

Reviewer 2 Report

The manuscript coherently describes a research project that explores the relationship between science and politics in the COVID-19 pandemic from the perspective of public health (PH) scientists in Germany with situation analysis. Great original work. I propose the following minor comments for the authors to consider.

Abstract : Add the titles of the sub-sections parts (3) and (4) of the summary.

Introduction : The introduction section is well written, reflecting the background of the study being reported. I recommend only one minor point, start each objective with a capital letter in a paragraph presented at lines 75-82.

Materials and methods: This section is well written. However, I would suggest adding the title study design before the first paragraph of materiel and method section.

Dissemination : I recommend adding the strengths and limitations of situational analysis.

Author Response

Dear reviewer,

we gratefully appreciate your effort and attention to our manuscript. Your comments were very helpful in revising the text. We address each comment point-to-point in the following.

Best regards.

Point 1:

Abstract : Add the titles of the sub-sections parts (3) and (4) of the summary.

Introduction : The introduction section is well written, reflecting the background of the study being reported. I recommend only one minor point, start each objective with a capital letter in a paragraph presented at lines 75-82.

Materials and methods: This section is well written. However, I would suggest adding the title study design before the first paragraph of materiel and method section.

Response 1

Thank you very much, we addressed the issues and modified the text accordingly.

Point 2:

Dissemination : I recommend adding the strengths and limitations of situational analysis.

Response 2

Thank you very much for this important comment. Strenghts of situational analysis include the focus on the situation (here the pandemic) as the unit of analysis and the attentiveness to its complexities by using SA‘s distinctive power analytics (social world/arenas and positional maps). This applies especially when studying implicated actors, e.g., actors silenced or marginalized. Furthermore, SA is a suitable framework for the development of interdisciplinary qualitative research, since SA mapping and analysis can be done collaboratively. Furthermore, we argue that it is challenging to capture the dynamics of the situation and how we will adapt our approach to this.  We have added corresponding considerations in section 2.2 (see line 284-295). Once the analysis has been completed, we will be able to reflect and discuss limitations and strengths of our methodological approach in more depth in our planned publications.

Following the editorial board's recommendation, we changed the section "Dissemination" to "Conclusion".
